# Urban Flood Modeling and Risk Assessment with Limited Observation Data: The Beijing Future Science City of China

**DOI:** 10.3390/ijerph20054640

**Published:** 2023-03-06

**Authors:** Huan Xu, Ying Wang, Xiaoran Fu, Dong Wang, Qinghua Luan

**Affiliations:** 1College of Water Conservancy and Hydropower, Hebei University of Engineering, Handan 056021, China; 2North China Municipal Engineering Design and Research Institute Co., Ltd., Tianjin 300074, China; 3National Institute of Natural Hazards, Ministry of Emergency Management of China, Beijing 100085, China; 4Hebei Provincial Research Center of Water Ecological Civilization & Social Governance, Handan 056021, China; 5Key Laboratory of Flood Disaster Prevention and Control of the Ministry of Emergency Management in China, Hohai University, Nanjing 210024, China

**Keywords:** MIKE URBAN, pipe network system digitalization, calibration, validation, risk of waterlogging

## Abstract

The frequency of urban storms has increased, influenced by the climate changing and urbanization, and the process of urban rainfall runoff has also changed, leading to severe urban waterlogging problems. Against this background, the risk of urban waterlogging was analyzed and assessed accurately, using an urban stormwater model as necessary. Most studies have used urban hydrological models to assess flood risk; however, due to limited flow pipeline data, the calibration and the validation of the models are difficult. This study applied the MIKE URBAN model to build a drainage system model in the Beijing Future Science City of China, where the discharge of pipelines was absent. Three methods, of empirical calibration, formula validation, and validation based on field investigation, were used to calibrate and validate the parameters of the model. After the empirical calibration, the relative error range between the simulated value and the measured value was verified by the formula as within 25%. The simulated runoff depth was consistent with a field survey verified by the method of validation based on field investigation, showing the model has good applicability in the study area. Then, the rainfall scenarios of different return periods were designed and simulated. Simulation results showed that, for the 10-year return period, there are overflow pipe sections in northern and southern regions, and the number of overflow pipe sections in the northern region is more than that in the southern region. For the 20-year return period and 50-year return period, the number of overflow pipe sections and nodes in the northern region increased, while for the 100-year return period, the number of overflow nodes both increased. With the increase in the rainfall return period, the pipe network load increased, the points and sections prone to accumulation and waterlogging increased, and the regional waterlogging risk increased. The southern region is prone to waterlogging because the pipeline network density is higher than that in the northern region and the terrain is low-lying. This study provides a reference for the establishment of rainwater drainage models in regions with similar database limitations and provides a technical reference for the calibration and validation of stormwater models that lack rainfall runoff data.

## 1. Introduction

Global climate change causes waterlogging in cities around the world. In 2021, a “7.20” extreme rainstorm occurred in Zhengzhou (China), a “7.13” extreme rainstorm occurred in Germany, and a severe storm also occurred in the northeast United States. Sudan and Pakistan have been hit by multiple rainstorms since June 2022, causing severe waterlogging disasters. Urban waterlogging poses a considerable threat to the natural environment, human life, and social economy [1,2,3,4]. China is one of the most flood-hit countries in the world due to monsoons [5]. On 20 July 2021, the “7.20” heavy rain event in Zhengzhou (Henan Province, China) affected 14,786,000 people and caused a direct economic loss of RMB 120.6 billion, resulting in considerable human and economic losses [6]. The aforementioned cases pertaining to emergency management in flood-prone areas imply that early warnings and risk assessments must be provided in a timely and effective manner [7]. Applicable regional urban stormwater models can simulate urban waterlogging and drainage pipe networks and evaluate the effectiveness of flood risk management schemes [8]. Model calibration and validation is a necessary process to ensure regional applicability. However, given the pipeline network and financial resource complexities, cities in many countries, including China, lack real-time monitoring of the flow of stormwater pipes, which hinders the calibration and validation of model parameters requiring real-time data while ensuring the regional applicability of the model. Knowing how to conduct model simulations in urban areas that lack monitoring data has become a difficult problem in storm flood risk management.

Different methods of model calibration can be applied to cities lacking monitoring data. The most common and basic method is the empirical calibration method of the rainfall comprehensive runoff coefficient. In 2009, the parameters of the Storm Water Management Model (SWMM) were calibrated with the runoff coefficient to provide new insights into the parameter calibration of the rainfall runoff model for areas lacking monitoring data [9]. This method can increase the amount of information about the calibration parameters and improve their estimation [10]. Since then, many scholars have used this method to calibrate and validate the parameters of urban stormwater models built in areas lacking data. Chen et al. used a hydrologic–hydrodynamic coupling numerical model to establish urban stormwater models in their study area, calibrated and verified the model parameters with the comprehensive runoff coefficient, and studied the effectiveness of runoff control on a previous pavement [11]. Peng et al. analyzed the performance of LID measurements under different design scenarios and performed comparative simulation and numerical modeling of the corrected comprehensive runoff coefficient to reduce the flood risk in their study area [12]. Hossain et al. and Wu et al. used the MIKE model and the comprehensive runoff coefficient method to realize the multidirectional coupling of drainage networks, urban surfaces, and river channels, and subsequently simulated the rainfall process and flood formation path in urban areas, comprehensively evaluated drainage systems, and forecasted the impact of extreme rainstorms and underground surface changes on urban drainage systems [13,14]. Hu used the runoff coefficient method to calibrate the main model parameters, simulate the control effect of different LID (Low Impact Development) measures on runoff of different rainfall events, and analyze the effect of rainfall flood control and utilization in an area whose pipe network system had not been built [15]. Zhang et al. used the runoff coefficient method to calibrate the SWMM model, analyze the feasibility of replacing the rainwater pipe network with a natural drainage system in a certain area, and determine the control effect of the single and combined LID schemes on runoff [16]. Zhang et al. built an urban stormwater model for a development zone lacking rainwater pipe network monitoring data, used the runoff coefficient method to calibrate the parameters, conducted a waterlogging security assessment in the study area, and comprehensively analyzed the flood risk of the main and surrounding plots [10].

As cities continuously develop, the types of underlying surfaces have become increasingly complex. For areas with a complex underlying surface, the maximum entropy algorithm has been successfully used in urban waterlogging risk assessment. Lin et al. developed a robust method for predicting future waterlogging-prone areas by coupling the maximum entropy (MAXENT) and the Future Land Use Simulation (FLUS) model. It was found that the proportion of impervious surfaces, population density, and proportion of green areas are key spatial drivers behind urban waterlogging issues. This method could support future urban design and waterlogging risk prevention [17]. However, certain errors have been observed in empirical estimations that use the runoff coefficient; these errors are usually caused by insufficient or inaccurate terrain measurements in numerical models or local terrain errors introduced by grid interpolation [11]. The runoff coefficient for model calibration may be combined with the validation of field survey data to improve the regional applicability of the aforementioned models and increase the accuracy of simulations; this combined approach also increases the simulation accuracy [18,19,20]. Hou et al. used the runoff coefficient method to calibrate and verify the measurement results of waterlogging risk points to assess the urban waterlogging risk in their study area. By combining geographic information technology and the SWMM model, they could visualize urban storm runoff under different return periods and evaluate the drainage capacity of a drainage network and the risk of urban waterlogging [21]. For highly dense urban areas, such as Yi Zhuang, Beijing (China), our team proposes the fine digitization of the underlying surface of an economic development core area. Then, the maximum runoff depth of typical points is taken as the validation factor, the field survey is taken as the main factor, and the runoff coefficient is used as a supplementary measurement, allowing us to complete the calibration and validation of the regional model. In addition, the response characteristics of production and confluence of different typical subcatchment areas and the risk of waterlogging traffic congestion are analyzed and evaluated by designing a scheme for heavy rainfall with different return periods [19].

Compared with the use of empirical calibration only, the combined calibration–validation approach with process measurement data can improve the regional applicability of the model. However, only the total runoff and confluence control parameters are considered (i.e., time and space are ignored) for the whole hydrological response process. For most urban stormwater models, the confluence process is based on hydraulic formulas, and many scholars have used them to calibrate and validate model parameters. Kumar Sarkar et al. established the MIKE URBAN model to simulate rainfall and drainage capacity under different rainfall return periods. They used the hydraulic formula to calibrate and validate their model, compared the simulated value with the calculated value, and found that the surface runoff increased gradually with an increasing return period. In their approach, problems of urban drainage and waterlogging were solved by drawing a submerged map and drainage density map based on the maximum flow value and DEM [22]. Vorobevskii et al. used the SWMM model and the SWSS system to analyze and investigate the relationship between regional rainstorm events and storm sewer system load, providing new insights into storm drainage system overload in urban areas. They used the Nash–Sutcliffe efficiency and Kling Gupta efficiency values of short-duration heavy rainfall events and the peak and volume error parameter performance formulas to calibrate and validate model accuracy according to a set of performance criteria [23]. To help urban dwellers effectively cope with rainstorms and floods, Zhang et al. established an urban waterlogging coupling model in newly built urban areas and simulated it in MIKE FLOOD. In their work, given the lack of hydrological measurement data, the pipeline flow self-test method was adopted to validate the rationality of the constructed model [24].

Stormwater model calibration and validation are important aspects of risk assessment of urban stormwater. In the process of such assessment, the flood discharge capacity of urban stormwater drainage pipes is a key factor [25]. In the above study, due to the limited flow pipeline data and measured rainfall data in the study area, there are certain errors only using the runoff coefficient to calibrate, which are caused by the insufficient terrain accuracy used in the numerical model or other elements. The methods of the formula validation and validation based on field investigation only consider the total runoff and confluence control parameters, which have limitations on the consideration of time and space for the whole hydrological response process. However, the accuracy and regional applicability of the model cannot be well guaranteed. On the basis of the research presented above, considering the influence of different factors and methods on model simulation, this study took the newly built urban area of the Beijing Future Science City of China as the study area and used three methods, namely, empirical calibration, formula testing and research validation, to calibrate and validate the model. Further improvements were made to the regional applicability of the model. Then, the pipeline filling degree was taken as the evaluation index, and points and sections that easily flood and accumulate water were analyzed to evaluate the waterlogging risk in the study area. The detailed discussion of the whole modeling process can provide a reference for regions with similar database limitations. Our work can be used as a reference for rainwater system modeling and waterlogging risk assessment in areas where relevant data are difficult to obtain or are lacking. The technical roadmap of the study is shown in the Figure 1 below:

## 2. Materials and Methods

### 2.1. Study Area

Beijing Future Science City of China, with an area of 10 km^2^, is located in the southeastern area of Changping District in the northern part of the Beijing Plain. The specific data sources and uses of this study are shown in Table 1.

The study area belongs to a newly built urban area and is dominated by technology enterprises located within a large space. Its surface and underground structures are complex, and its underlying surface data are difficult to obtain. Furthermore, ground data and observations are relatively insufficient. The study area is divided by the Wenyu River and Dingsi Road (Zhengfu Street) into southern and northern areas. The northern area is located in the southeastern part of Xiaotangshan Town, with an area of 2.19 km^2^, while the southern area is located in the eastern part of Beiqijia Town, with an area of 4.46 km^2^. The green landscape between the two districts totals 3.38 km^2^. According to the investigation of the catchment area covered by the rainwater pipe network of the two districts, the river and the bank space have zero release; this aspect is not included in this research. This plan is for the Beijing Future Science City of China to be constructed into areas for research of central enterprises and creative service facilities. Its land use includes zones for public facilities, water, multifunctional land use, green space, residential land, roads and squares, and municipal public facilities of which the precipitation area is shown in Table 2.

The Wenyu River crosses the area, and the area is quite flat. The annual precipitation and interannual precipitation distribution is quite uneven and concentrated in June to September at 78.3% of the total annual precipitation, in which the precipitation in July is the highest (32.7% of the whole year). In the presence of extreme precipitation, the water level of the upstream area easily rises, and the river can hardly accommodate the rainfall in the area; subsequently, waterlogging occurs. From the risk profile of the historical flood and waterlogging disasters in Beijing, the Beijing Future Science City of China is in a high-frequency risk area.

### 2.2. Model Selection

In the process of urbanization, the underlying surface conditions and stormwater pipe network system are increasingly complex and changeable. Urban stormwater models, which can reasonably and accurately simulate the urban hydrological process and analyze the risk of urban floods and waterlogging, are widely used at home and abroad. Common urban storm flood models include the InfoWorks CS distributed model developed by the British Wallingford Software Company [26], the STORM model developed by the U.S. Army Corps of Engineers [27], the SWMM model developed by U.S. Environmental Protection Agency [28], a series of MIKE models developed by the Danish Institute of Hydrodynamics [29], etc. Some scholars have used the SWMM model to simulate the impact of urbanization on runoff [30] and analyzed the impact of the infiltration capacity of the osmotic zone on rainwater runoff [31], providing a theoretical basis for improving the infiltration capacity of the osmotic zone and effectively controlling urban waterlogging disasters. In 2014, Wu et al. used MIKE URBAN and MIKE 11 in the MIKE FLOOD platform for coupling for the first time in Wuhan, China, proving that this model is an effective tool for urban rainwater system assessment [32]. Similar studies have been conducted in other parts of China, such as the simulation of regional urban floods in Lujiazui, Shanghai [33]. A flood simulation was conducted in Jinan, China [34].

MIKE URBAN is a widely used drainage network simulation software. It was developed by the Danish Hydraulic Institute (DHI), integrated with ESRI’s ArcGIS, System CS, and the water supply network WD, and the calculation models use the surface flow, open channel flow, pipeline flow, water quality and sediment transport in urban launching system approaches. This gives it a strong ability of urban water cycle and associated process simulation [35] as a computer program that can model the static and delayed hydraulic and water quality characteristics, with Modeling of Urban Sewer (MOUSE) and SWMM two drainage calculation engines. The MOUSE engine is the urban storm runoff model developed by DHI in 1984. The main modules include the rainfall infiltration module, surface runoff module, pipe flow module, real-time control module and sediment transport and water quality module. Limited by the 1D simulation of storm runoff, water quality and sediment transport in the urban drainage system, the model is always coupled with the 2D hydrodynamic model MIKE21 in MIKE FLOOD as the platform and linked with the 1D river network hydrodynamic model MIKE11 to establish a 1D and 2D coupled urban flood model to describe the urban drainage pipe network, open channel, drainage channel, a variety of hydraulic structures and 2D slope flow, river basin floods and urban floods [36]. Furthermore, this software can predict the hydraulic deficiencies, overflow sites, flood inundation areas, and effect of real-time control [37]. In contrast to SWMM and InfoWorks CS, the MIKE URBAN model has the advantages of requiring less data, simple modeling and operation, fewer parameters and more accurate results [38]. This model is suitable for areas where precipitation and runoff data are scarce and parameter evaluation is difficult.

The modeling data of the study area are limited to the basic data of the pipe network system, regional planning text data and field survey data, and the basis of modeling data is relatively weak, which hinders the construction of a 2D surface submergence model. Therefore, the MOUSE module in the MIKE URBAN model was selected as the tool for simulation in this study. In addition, MIKE URBAN is suitable for any type of free surface flow, and pipeline pressure flow alternately changes the pipe network [39], which helps to analyze the results of pipe flow with respect to the pipeline filling degree. Apart from the selected module mentioned above, the pipe flow simulation results were displayed visually on the ArcGIS platform, and the bottleneck of the pipe network system was analyzed.

### 2.3. Pipe Network System and Subcatchments

The rainwater drainage system model of the Beijing Future Science City of China was established according to the rainwater drainage profile and rainwater management layout through the MOUSE module of the MIKE URBAN software. Related data were presented in the “Beijing Future Science Park (southern) Rain Sewage Exclusion Planning (modified version), Beijing Future Science Park External Rain Sewage Drainage Planning and Beijing Future Science City Rainwater Control Special Planning”, by the Beijing Water Science and Technology Institute (BWSTI). In addition, the pipe network model data were checked through the project inspection tool before the model was established in order to ensure the correctness of the results, such as the pipe network and elevation data [40].

As the land use types of the study area are mostly large-scale areas, such as the central industrial and residential areas, the confluence direction and node of subcatchment areas were difficult to set. By referring to the field investigation, considering the layout of rainwater control and utilization, distribution and the related hydraulic connections of rainwater wells and rainwater pipes in the surrounding roads of the construction area, the pipe networks were digitalized. The subcatchment and confluence rainwater wells were set via the following steps.

(1)Pipe network digitization

The rainwater pipe network system was partitioned. Without considering the interaction of the surface 2D flow, the rainwater pipe network partition boundary was taken as the virtual watershed boundary line and the definition of partition control. The red line in Figure 2 lays the foundation for the fine division of the subcatchment area.

The direction of the pipe network and the distribution of the pipe outlet were determined, and the regional rainwater pipe network system was partitioned according to the rainwater control layout in consideration of the surface confluence. In particular, the southern rainwater pipe network was divided into four systems (A, B, C and D), whereas the northern rainwater pipe network was divided into three systems (E, F and G). The rainwater in A and B was discharged into the Wenyu River and its road; the rainwater in C and D was discharged into the Lutuanxi Trench; the rainwater in E was discharged into the Wenyu River; the rainwater in F was discharged into the earthen drainage channel; and the rainwater in G was discharged into the present passing culverts of the Jing-Cheng expressway.

The project test tool was used to check the topology of the pipe network and the connection between catchments, and the model was adjusted according to the results of the inspection. This scheme could ensure the correctness and stability of the model. The topology of the rainwater removal system of the Beijing Future Science City of China is shown in Figure 3.

(2)Subcatchment division based on zoning

(a) For the area with a sparse distribution of pipe sections (northern region), the Thiessen polygon method was used to divide the subcatchment area, and the findings were combined with the confluence production and drainage information obtained from the field investigation; then, local adjustment was performed. The confluence node was selected as the rainwater well closest to the center of the subcatchment area.

(b) For the area with dense pipe sections (southern area), the subcatchment areas were divided into detailed data according to the ground elevation and field investigation information, and they were combined with the pipeline direction, street and distribution data of the buildings in the area and the actual drainage data obtained from the investigation. On this basis, the confluence rainwater wells and the confluence direction of the subcatchment areas were specified.

In other words, by generalizing and constructing the model, the rainwater drainage wells could be assumed to be the nodes of the rainwater drainage system. The nodes numbered 50 in the northern area and 154 in the southern area, with a total of 204 rainwater ports. Among them, the northern pipe network had three outlets (E6, F6 and G4), whereas the southern pipe network had four outlets (A8, B14, C6 and D16). The rainwater pipeline covered 47 sections of rainwater pipes on the north (pipeline length: 13,060 m) and 150 sections on the south (pipeline length: 33,240 m), with a total pipeline length of 46,300 m. The area was partitioned into 91 subcatchments according to the partition results and the planned layout of rainwater control and utilization. The northern area was divided into 32 subcatchments (catchment area: 439.93 ha), whereas the southern area was divided into 59 subcatchments (catchment area: 214.60 ha), totaling 650 ha of water area, which was almost all the parts covered by rainwater pipelines, except untapped land, river water surfaces and riverside green space, which do not drain water. The results of the rainwater drainage system generalization for the Beijing Future Science City of China and the catchment area partitioning are shown in Table 3.

### 2.4. Methods of Parameter Calibration and Validation

The main purpose of calibrating and validating model parameters is to improve the accuracy of the model, minimize the relative error between the simulated and actual values, and improve the degree of fitting between them [14]. Here, the parameters of the validation points with large errors between the simulated and actual values were constantly adjusted until the simulated values of all validation points were closest to the measured values, implying improved simulation accuracy of the model.

In the process of parameter evaluation and validation, the model parameters were divided into total amount control parameters (mainly by considering losses of all types) and confluence control parameters. The runoff model was established using the time/area (TA) curve model. The total amount control parameters included the impermeable area ratio (%), initial loss (mm) and hydraulic damping coefficient, while the confluence control parameters included the catch time TC (min) and confluence TA curve model. The flow model was established based on the dynamic wave formulas given by the Saint-Venant equations, with the pipeline Manning coefficient taken as the main influencing parameter. 

Given the observed data limitations, this study utilized MIKE URBAN, which requires few parameters and is easy to estimate. The simple TA curve model was used to calculate the runoff, while the Saint-Venant equations were used to calculate the pipeline flow. As the newly built urban area lacked measured rainfall runoff data, the measured “21 July 2012” rainstorm data of Beijing were initially used, followed by the comprehensive runoff coefficient, as the validation target to evaluate the model parameters [41]. Then, the pipeline hydraulic calculation formula was used to validate the simulated maximum flow rate at the outlet, allowing the model parameters to be further adjusted. Finally, the runoff water depths from the simulation were compared with the typical validation point waterlogging water depth obtained from the field investigation to adjust and determine the parameters. The reliability of the model was verified by modeling the Beijing “23 June 2011” rainstorm (data were provided by BWSTI).

#### 2.4.1. Empirical Calibration

The runoff coefficient is usually calculated based on the weighted average of the land use types, from which the average runoff coefficient of a catchment area can be determined. The comprehensive runoff coefficient of urban area can be determined by consulting the outdoor drainage design manual (GB 50014-2021); the value of the integrated runoff coefficient for complex areas is shown in Table 4 [42].

#### 2.4.2. Formula Validation

During the parameter evaluation, given the lack of underlying surface data for the subcatchments, the internal condition of the subcatchment was also insufficiently described, but the total runoff and confluence control parameters were set. Furthermore, information about the whole hydrological response process in terms of time and space was insufficient. While ignoring the influence of human activities and micro terrain on the hydrological process implies faster simulation, they may also lead to inaccurate results (e.g., shorter confluence time, larger runoff, larger quantity of confluence rate, and larger flow rate at the outlet). The aforementioned bias was alleviated by using the pipeline hydraulic formula and the model results, allowing for the validation of the maximum flow rate of the pipeline outlet and other model parameters.

Cross-sectional area *A* and hydraulic radius *R* were obtained by the linear interpolation method based on the simulated results of the maximum fullness of the pipe section and the hydraulic factors of circular pipes with different filling degrees (Table 5). In particular, pipe diameter *d*, bottom slope *i*, and roughness *n* were determined according to the actual investigation condition of the pipeline, upstream and downstream elevation, and material of the pipe, respectively. Then, cross-sectional area *A* and hydraulic radius *R* were calculated according to Equations (1) and (2). Then, the hydraulic calculation formula of the pipe free flow (Equation (3)) was used to adjust the maximum simulated flow rate of the connecting section of the main pipe.
(1)A=bh
(2)$$R=\frac{bh}{b+2h}$$
(3)$$Q=AC\sqrt{Ri}\;or\;Q=R^{2/3}i^{1/2}$$
where *b* is the rectangular cross section width (m); *h* is the water surface depth (m); *Q* is the outlet maximum flow rate (m^3^/s); *A* is the cross-sectional area (m^2^); *R* is the hydraulic radius (m); *i* is the bottom slope; and n is the pipe roughness [43].

#### 2.4.3. Validation Based on Field Investigation

In other studies, the validation is processed by comparing the measured value of the rainfall runoff process with the simulated value [44]. Due to the lack of measured runoff data, this study compared the observed maximum submerged depths of typical sites with the simulated maximum runoff depth and then estimated the process of the model parameters. A comparison of the maximum submerged depth and runoff depth of the validation point indicated that the model results could meet the accuracy requirements and achieve the goal of parameter validation. Thus, on the basis of the “21 July 2012” rainstorm simulation, combined with the actual field investigation data of the rainfall process, the water depth of the typical location was compared and verified, and an accurate adjustment of the model parameters was eventually completed.

The main points of the survey and validation included surface convergence nodes, road intersections, pipe convergence points, and points of low ground elevation and outlets. The principle of validation point screening was applied as follows. (1) In line with the concept of controlling drainage by partitioning, combined with the data of the ground elevation of rainwater wells from “Beijing Future Science Park Urban Rainwater Control Special Planning”, this research first determined the low points of the ground where waterlogging would likely easily occur by analyzing rainwater ground elevation data. (2) On the basis of the simulated “21 July 2012” rainstorm, the pipe section of the high load degree was determined. (3) In accordance with the actual research conditions, validation points were selected. This method could reduce the blindness of the validation point in the actual research process, ease the parameterization of the model parameter, improve the efficiency of the parameter validation, and establish a good foundation for realizing the evaluation and validation of the model parameters. The final validation points are shown in Figure 4.

The MIKE model calibrated and verified by combining the above three methods had better regional applicability; we then used the model to simulate the load of the pipe network in the northern region and southern region under different rainfall scenarios in different return periods.

### 2.5. Scenario Design and Assessment of Waterloging

#### 2.5.1. Scenario Design

According to the “Beijing Hydrologic Manual-Heavy Rains Atlas [45] and the Runoff Computing Standards of Beijing Urban Rainwater Drainage System Planning and Design (DB11/T 969-2016)” [46], Beijing can be divided into two rainstorm zones, District I and District II. Each rainstorm zone is based on the township level. Beijing Future Science City of China is located in District II of the Beijing Rainstorm Zone. The intensity of the designed rainstorm in Zone II is calculated according to Equations (4) and (5).
(4)$$q=\frac{591\left(1+0.893lgP\right)}{{\left(t+1.859\right)}^{0.436}}$$

Scope of application: 1 min ≤ t ≤ 5 min, P = 2a − 100a
(5)$$q=\frac{1602\left(1+1.037lgP\right)}{{\left(t+11.593\right)}^{0.681}}$$

Scope of application: 5 min ≤ t ≤ 1440 min, P = 2 – 100a.

Where *q* is the intensity of the designed rainstorm (L/(s·hm^2^)); *t* is the duration of the designed rainfall (min); *p* is the designed return period (year).

According to the data of rainstorm waterlogging disasters from 21 July 2012 up to the present [47], the duration of the most disastrous rainstorm was approximately 24 h. Therefore, in this research, the designed rainfall process under different return periods was calculated according to 10-year, 20-year, 50-year and 100-year return periods and the rainfall duration was set to 24 h. The designed rainfall process was computed based on the different return periods, and the MIKE Zero software was used to process data and generate the “*.dfs0” file to meet the requirements for model input [40,43]. By using the selected information as the rainfall zone boundary for the rain drainage system model, a simulation analysis of the designed storm flood for Beijing Future Science Park and a risk assessment of urban waterlogging could be jointly conducted. The designed rainfall process is shown in Figure 5, and the designed rainfall under different rainfall return periods is shown in Table 6.

#### 2.5.2. Assessment Index and Principle for Urban Waterlogging Risk

The critical rain drainage system and road spots and the sections prone to waterlogging in the Beijing Future Science City of China were determined via scenario simulation. This study analyzed the bottlenecks of rain drainage systems and road sections based on the simulated result of the pipeline filling level. The pipe filling was calculated as the depth divided by the pipe height; e.g., if the pipe is running under pressure, then the ratio is greater than 1.0 [48]. For the given design flow, the ratio of the rainwater water depth *h* in the pipeline to the pipe diameter *D* is called the design degree of filling (or the water depth ratio) *a*, which is given by
*a* = *h/D*(6)

In this study, “*a*” indicates the degree of pipeline filling. When a < 1, the pipeline flow state is an open channel flow. When *a* = 1, it is called full flow. When a > 1, the pipeline flow is under pressure. In other words, h is not the pipeline’s actual water depth, as the pipeline in a state of high pipe head pressure (i.e., as the pressure inside the pipe increases, the head of the pressure measurement tube also increases, and the value also increases). In cases of *a* < 1, the network is in a state of safe operation, and the risk of waterlogging is not apparent. In cases of *a* > 1, the network is in a state of full-load operation. As rainfall continues, the drainage capability of the networks will likely meet its bottleneck, and the problem of overflow and waterlogging is apparent. In cases of 1 < *a* < 5, the open channel flow changes into pipe flow, and some networks are at full capacity. The risk of urban waterlogging exists, but it is likely manageable because waterlogging will decrease once the rainfall ends. In cases of 5 < *a* < 10, most networks are at full load. Overflowing in jointed nodes, waterlogging and urban floods, and bottlenecks in the drainage system will likely occur. The long-standing rainfall is beyond the drainage capability, and the risk of urban waterlogging is comparatively high. In cases of 10 < *a*, the whole drainage system loses its capacity of discharge. Flooding in road sections and of an increasing number of drainage wells with overflows will likely occur, and the risk of urban waterlogging in certain areas is high. The risk becomes even higher as rainfall continues. In such situations, moveable pumping stations are needed to accelerate flood drainage. The waterlogging risks of different pipe filling degrees are shown in Table 7.

Filling level and duration are parameters that can directly reflect the overloaded operation of pipe networks and indirectly reflect the potential risk of urban waterlogging. As the peak value of flow velocity rises, most networks move towards a state of overloaded operation. When the pipe diameter of a rain spout is large, its drainage capability is strong. Thus, the filling level can be regarded as *a* < 1 only at the end of the drainage pipe near the rain spout.

## 3. Results

### 3.1. Model Calibration and Validation

In the process of empirical calibration, MIKE Zero was used to process the observed precipitation data of the “21 July 2012” heavy rain in Beijing, the rainfall process is shown in Figure 6, and generate a 721.dfs0 rainfall input file [49]. The precipitation of the rainstorm was 264.5 mm, and the rainfall lasted for 26 h; these data were taken as the boundary conditions of the model rainfall. The comprehensive runoff coefficient of the whole study area was 0.404, which was determined by weight-averaging the runoff coefficients of the subcatchments and calculating the model by loading the accumulated precipitation. According to the planning scheme and the field investigation, the impervious area was approximately 45% of the total area, and the corresponding comprehensive runoff coefficient ranged from 0.40 to 0.60, which is consistent with the results of the model. The results of the runoff coefficient with respect to typical rainfall in the study area are shown in Table 8.

After the empirical calibration of the model parameters, the pipe flow in the northern and southern regions was simulated. It was found that the relative error between the simulated value and the calculated value at some outlets was large, which indicates that empirical calibration only cannot guarantee the high accuracy of the model. Then, the hydraulic Formulas (1)–(3) were used to adjust the maximum simulated flow rate of the connecting section of the main pipe in the study area by adjusting the model parameters, so that the relative error between the simulated value and the calculated value was kept within the specified range. The calculation results validate the suitability of the model parameters. The adjustment of the maximum simulated flow rate of the main outlet connection section is shown in Table 9.

The relative error of the pipeline flow can be reduced to less than 25% (i.e., the allowable range) according to the hydraulic formula, implying optimization of the model parameters. In the absence of field-measured flow data, self-optimization of parameters can reduce the bias between the simulation results and the actual condition and improve the accuracy of the model and the applicability of the model parameters.

In addition, the method of validation based on field investigation is adopted to further improve the accuracy of the model which have been calibrated and verified. On the basis of consultation with local residents and the investigated trends of flood traces for understanding the local impact of the “21 July 2012” heavy rain on the region, this study did not find any obvious waterlogging occurring in the urban area, and the depth of the waterlogging was between 10 and 20 cm, which is consistent with the simulated maximum runoff depth of 12 cm. Thus, the set of model parameters was reasonable. The calibrated and verified parameters of the model are shown in Table 10.

Combined with the validation results mentioned above, the model was further confirmed by analyzing the storm runoff process of “23 June 2011” in Beijing, as shown in Figure 7. The simulated value of the rainfall process was consistent with the calculated value. According to field investigation and local resident consultation, this rainfall had a small impact on the area, with no obvious waterlogging on the road. Under typical rainfall conditions, the comprehensive runoff coefficient of the model simulation was 0.433, as shown in Table 5. The average runoff depth was 5.24 cm, calculated according to the runoff coefficient and the accumulated rainfall in the subcatchment area. Clearly, the pipe network in the study area was not affected by drainage capacity. The simulation results are in accordance with the investigation results of the validation point, proving the reliability of the model.

### 3.2. Flooding Risk Assessment under Different Rainfall Return Periods

As shown in Figure 8, the influence of the designed rainfall process on the rain drainage system in the northern region can be simultaneously learned for different return periods. The selection of a particular moment is based on the following principle: (1) the moment when the peak value of the network flow velocity appears and (2) the moment after the rainfall peak value appears. However, peak values do not appear at the same time for different networks. Here, the moment when peak values of flow velocities appear in most networks was selected as the moment of greatest risk of flooding.

Figure 9 shows the influence of the designed rainfall process on the drainage system in the southern region under different design return periods. On the basis of the changing filling level caused by the sudden shift in pipe diameters and the duration of the rainfall’s influence on pipelines, the moment with the highest filling level and maximum load is selected from the simulated result.

The selected moment in the southern region appears relatively earlier than that in the northern region. The reasons can be summarized as follows. For rainfall with a 100-year return period, the drainage system is under overloaded operation after the first peak value of rainfall intensity appears because of the large amount of rainwater and its high intensity. The drainage system cannot bear long-standing rainfall, further suggesting that the whole southern region will likely face serious waterlogging disasters.

On the basis of the simulated results of the designed rainfall process’s influence on the drainage pipeline filling level in the northern region for different design return periods, the number of drainage pipelines and jointed nodes and the pipe flow states were used to represent the condition of the whole drainage system. Here, repeat counting was conducted for two filling levels. The simulated data are shown in Table 11.

In order to provide technical guidance for the regional management of the Beijing Future Science City of China, we compared and analyzed the waterlogging risks of different pipe network densities, and drainage capacity and terrain areas in the northern region and southern region.

(1) In terms of selecting the same moment, in the 10-year, 20-year, 50-year and 100-year return periods the number of pipelines under a state of overloaded operation also increases. The length of affected pipelines, the number of overflow wells and the area at risk of overflow and waterlogging also increase. For different moments, as the return period increases, the duration of overloaded operation becomes longer, and the duration for the rain flow to recede and the waterlogging risk are even longer.

(2) In terms of the difference between the northern and southern regions, the following points can be inferred: (a) Given the same rainfall conditions, when the return period is relatively short, more open channels flow in the southern region, and the risk of waterlogging is lower than that in the northern region. This situation is related to the pipeline network density (i.e., the pipeline network density in the southern region is higher than that in the northern region) and the drainage capability. (b) Given the same rainfall conditions, when the return period is relatively long, the number of pipelines under overloaded operation increases in the southern region, and the risk of overflow and waterlogging increases. The situation is related to the terrain of the southern region, which is low-lying and prone to waterlogging.

(3) In terms of the whole drainage system, when the return period is relatively long, the state of overloaded operation is more serious, and a bottleneck of drainage ability emerges. For example, sections E5-5 to E5-6 in the northern region and sections B7-1 to B7-2 in the southern region are always overloaded. As the return period increases, the impact on the pipeline worsens. The bottleneck lies in the difficulty of meeting the need for a designed rainfall with a longer return period.

## 4. Discussion

In this study, the methods of empirical calibration, formula validation and validation based on field investigation were used to calibrate and validate the parameters of the model in the area with limited observation data. It improved the accuracy of the model and the regional applicability of the model. During the validation of the hydraulic formula, it was found that for the rectangular outlet pipe section of drainage systems D and E, the relative error was large, and adjustment was insufficient to reduce the error. For example, taking the hydraulic formula of the pipeline, the formula validation clearly entailed defects, as it used the actual roughness parameter of the pipe as the sensitive parameter, hence the underestimation. In addition, drainage systems D and E had many pipes and long pipe lengths, and the actual roughness evaluation was also underestimated, further resulting in shorter pipe convergence times and large fluctuations in the peak flow at the outlet. Moreover, the control area was large, and the export section had many sinks and sources, coupled with the insufficient depiction of the underlying surface of the subcatchment area. These factors jointly resulted in the large peak flow of the export cross-section, and they were difficult to control. The proposed method can improve the applicability of the model parameters for areas with scarce pipeline flow data. In the future, if more pipeline information is available, then the model simulation accuracy and the applicability of the parameters can be further improved.

During the field survey, due to the limitations of time and manpower, this study selected eight main points of the survey and validation according to the principle of validation point screening. On the basis of consultation with local residents and the investigated trends of flood traces, it found that the depth of the waterlogging was consistent with the simulated maximum runoff depth. However, if the number of survey points can be increased in the future, the error of actual value and simulated value can be further reduced, and the regional applicability of the model can be better improved. Moreover, the subjectivity of interviewing residents must be considered: they are greatly affected by the outside world. Additionally, with the development of the era of big data, the monitoring technology of road ponding is constantly updated. In the future, we can make full use of video monitoring equipment, properly encrypt the video monitoring points of waterlogging, and improve the accuracy and timeliness of waterlogging information [50].

This study aimed to improve the accuracy and regional applicability of the model by combining different methods of model calibration and validation. In recent years, there have been many new methods developed for calibration and validation. The emergence of user-generated content (UGC) provides new opportunities for flood risk management. The new research showed that the proposed historical UGC-based model is practical and has good flood risk-mapping performance [51]. At the same time, remote sensing technology can also be applied to calibration and validation of models. Mason et al. used a TerraSAR-X image of a one-in-150-year flood near Tewkesbury, in the United Kingdom, in 2007, for which contemporaneous aerial photography exists for validation. The TerraSAR-X radar satellite can accurately extract the visible area and sheltered area of urban waterlogging of a water body, which can be used to calibrate and validate the urban storm waterlogging model [52]. Dwivedi et al. used meteorological and remote sensing data combined with geomorphological and geological information to assess the hydrometeorological disasters risk. Landslide susceptibility was analyzed by the maximum entropy model (MaxEnt). Rainfall-induced flash flood conditioning factors were classified and ranked using a weighted overlay approach to draw a flash flood risk map [53]. There are certain limitations in the calibration and validation of a model by a single method, and the accuracy of the model cannot be guaranteed. In the future, various methods can be combined in different ways to calibrate and validate the urban flood model, so as to minimize the limitations of each method, reduce the error between the simulated value and the actual value of the model, and improve the accuracy and regional applicability of the model.

## 5. Conclusions

This research took the Beijing Future Science City of China as its study area and used the MIKE URBAN software to build a rainwater removal system model. Furthermore, this research considered a situation in which rainfall data were lacking and the simulation accuracy of the model was difficult to guarantee (i.e., only two measured rainfall data points were available). On this basis, a new method of model calibration and validation is proposed, combining the three methods of empirical calibration, formula calibration, and survey calibration. First, the comprehensive runoff coefficient method was used to empirically calibrate the model parameters. Then, the hydraulic formula method was used to test the simulation results of the maximum discharge of the drainage outlet. Finally, the water depth data and the simulated runoff depth results were obtained by investigation and then checked, and calibration work of the system model parameters was completed to ensure the applicability of the model parameters and the stability of the model. Subsequently, a scenario simulation of the designed rainfall process under different return periods was conducted, the simulation results of the pipeline filling degree were analyzed, and the points and sections prone to flooding and accumulation in the study area were determined. The following conclusions can be drawn:

(1) In this study, first of all, the runoff comprehensive coefficient was used to calibrate the model parameters, so the comprehensive runoff coefficient of the whole study area was consistent with the corresponding comprehensive runoff coefficient. Then, the pipe flow in the study area was simulated. It was found that the relative error between the simulated value and the calculated value at some outlets was large, indicating that empirical calibration alone cannot guarantee the accuracy of the model. Then, the hydraulic formulas were used to adjust the maximum simulated flow rate of the connecting section of the main pipe in the study area by adjusting the model parameters, so that the relative error between the simulated value and the calculated value was kept within 25%. Finally, the method of validation based on field investigation was adopted to further improve the accuracy of the model, which had been calibrated and verified. On the basis of consultation with local residents and the investigated trends of flood traces for understanding the local impact of the “21 July 2012” heavy rain on the region, it was found that the depth of the waterlogging was consistent with the simulated maximum runoff depth. Thus, the set of model parameters is reasonable. Combining three methods of empirical calibration, formula calibration, and survey calibration to calibrate and validate the model, which could more accurately adjust the model parameters and improve the model accuracy, allowed the constructed model to be more suitable for the study area. This hybrid method is feasible and applicable for areas lacking measured runoff data. This research can provide a reference for the establishment of rainwater removal models in areas with the same data limitations.

(2) In terms of the model design, as rainstorms continuously recur (i.e., the return period increases), the load of nodes and pipelines in the study area, the number of full flow pipe sections and the number of high-load pipe sections all tend to increase. Furthermore, the high-load pipe segment is correlated with the overflow node, and a composite appears in certain areas. When the pipeline is under high-load operation, the drainage capacity is limited, and the node water level gradually increases. When the node water level is higher than the ground elevation, overflowing tends to occur, resulting in a continuous increase in the pipeline load. In addition, overflow nodes and high-load pipe sections are mostly found at the joint of the branch pipe and main pipe. The drainage capacity of the branch pipe is insufficient because of the larger amount of surface water being drained into the catchment area, resulting in an increased pipeline load.

(3) The drainage capacity of the study area can meet the design standard of a single rainstorm in 10 years. However, the risk of waterlogging is still high. For the southern region, the risk of waterlogging is relatively low.

## Figures and Tables

**Figure 1 ijerph-20-04640-f001:**
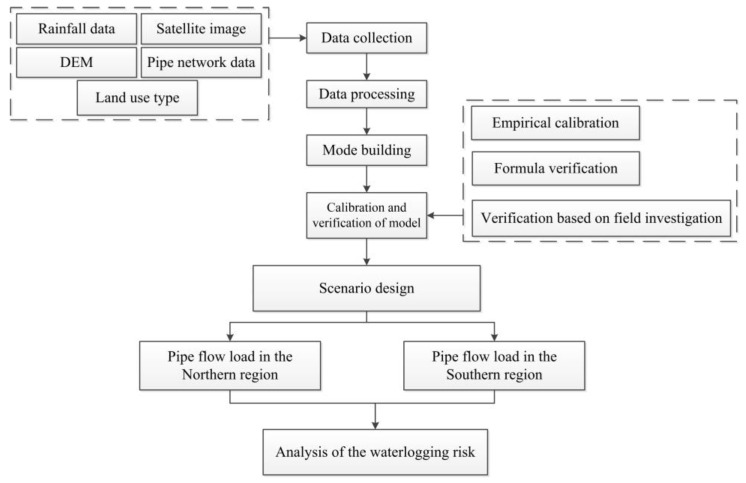
Technical roadmap.

**Figure 2 ijerph-20-04640-f002:**
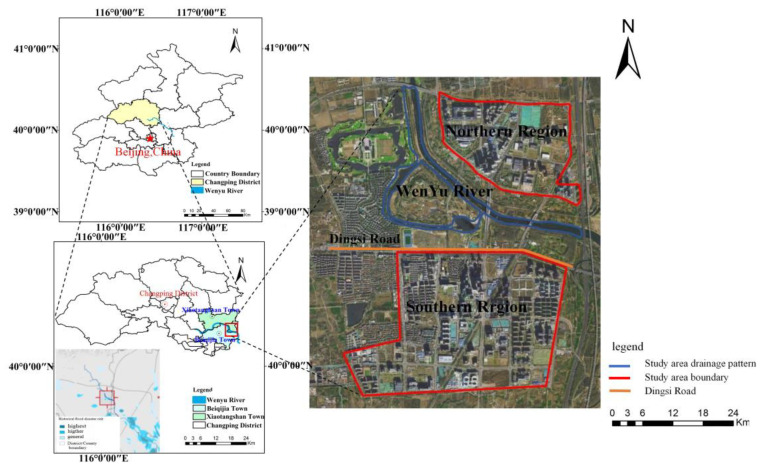
Location of the research area.

**Figure 3 ijerph-20-04640-f003:**
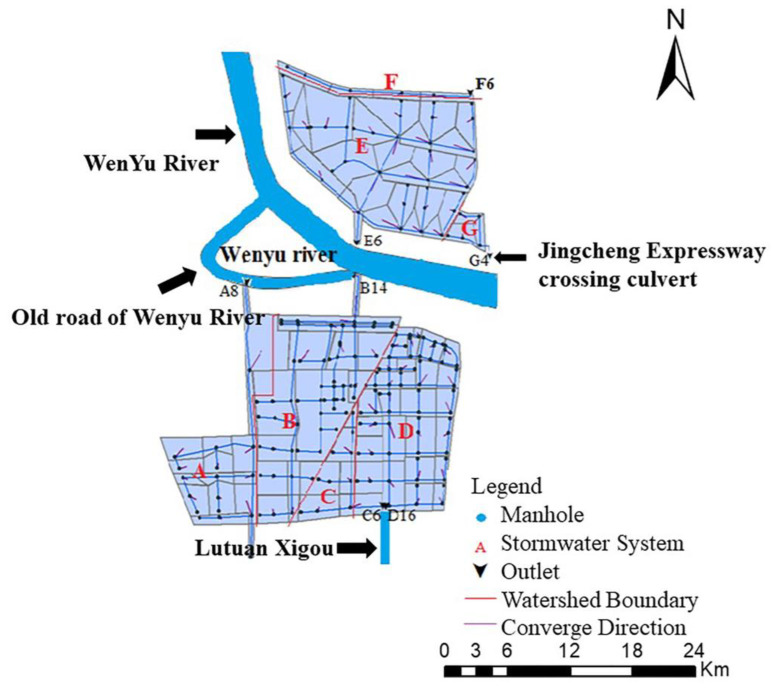
Overview of the rainwater drainage system of the Beijing Future Science City of China (A–G are the divided rainwater pipe network systems).

**Figure 4 ijerph-20-04640-f004:**
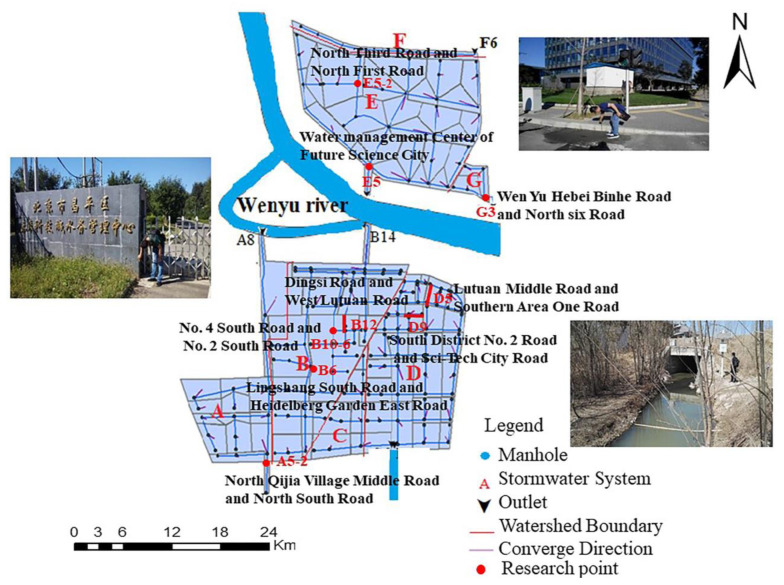
Validation points (A–G are the divided rainwater pipe network systems; A5-6, B6, B12, B10-6, D5, D9, E9, E5-2 and G3 are the final validation points).

**Figure 5 ijerph-20-04640-f005:**
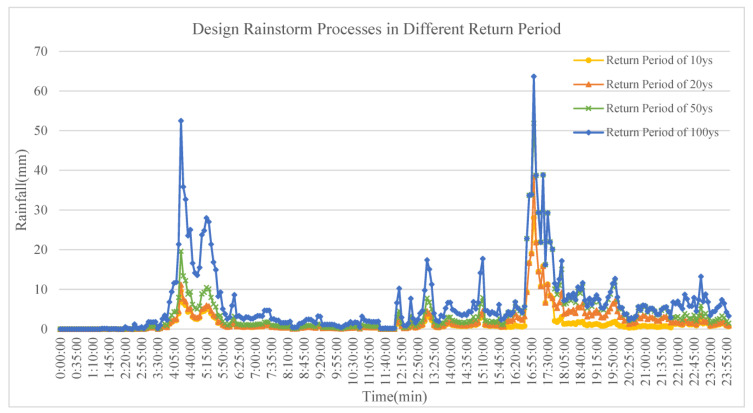
Designed rainfall process under different design return periods.

**Figure 6 ijerph-20-04640-f006:**
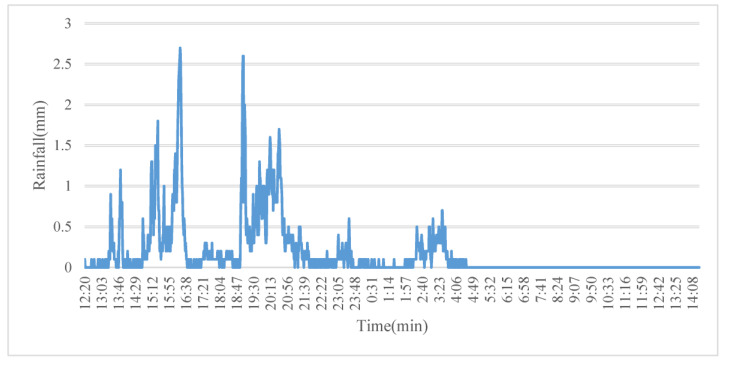
Typical measured “21 July 2012” rainstorm in Beijing.

**Figure 7 ijerph-20-04640-f007:**
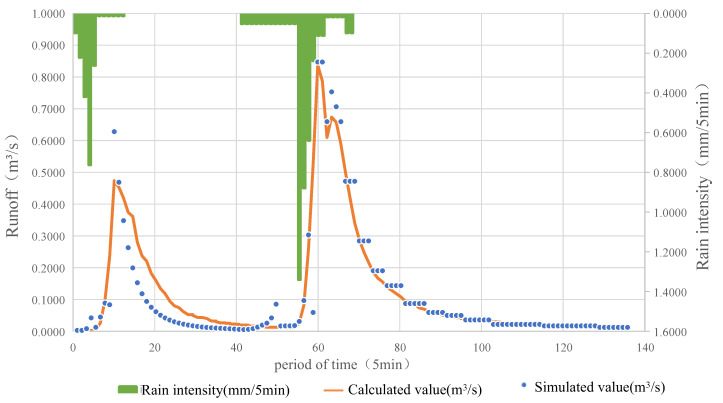
Validation result of “2011 June 23” rainstorm in Beijing.

**Figure 8 ijerph-20-04640-f008:**
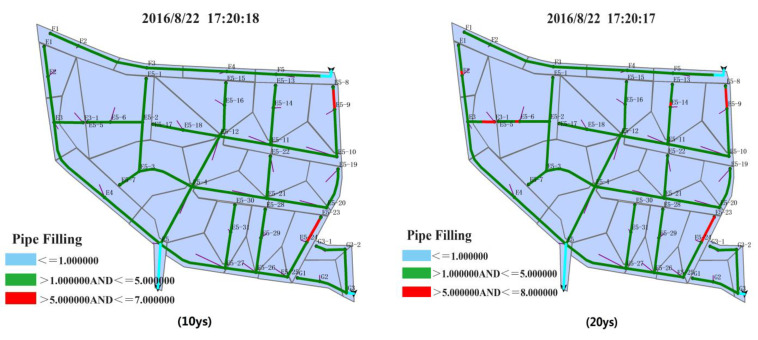
Simulation results of the influence of the designed rainfall process on the pipeline filling degree in the northern region under different design return periods (E1–G3 are the numbers of the pipes and nodes).

**Figure 9 ijerph-20-04640-f009:**
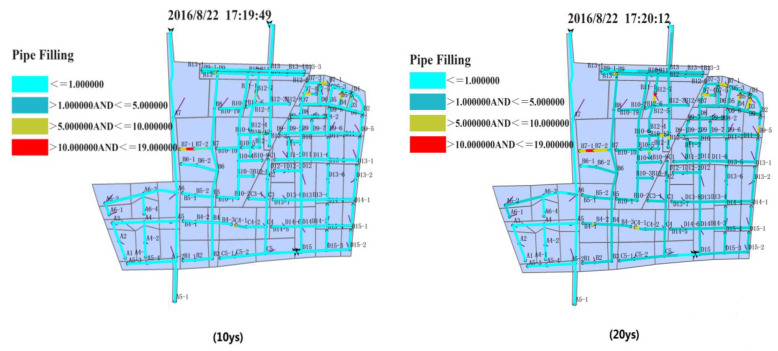
Simulation results of the influence of the designed rainfall process on the pipeline filling degree in the southern region under different design return periods (A1–D15 are the numbers of the pipes and nodes).

**Table 1 ijerph-20-04640-t001:** Requirements and usage of data.

Data	Source	Usage
DEM	Google Maps	Regional topographic reference and watershed division
Pipe network data	Beijing Water Science and Technology Institute	Establish the relationship model of runoff generation and concentration in drainage process
Rainfall data		Simulate model rainfall process
Underlying surface data		Extraction land type
Land use planning map	Beijing Future Science Park (southern) Rain Sewage Exclusion Planning (modified version); Beijing Future Science Park External Rain Sewage Drainage Planning; Beijing Future Science City Rainwater Control Special Planning	Division and parameter setting of catchment area in planning model
Topographic map of the planned area		Regional terrain reference and watershed division of planning model

**Table 2 ijerph-20-04640-t002:** Land type and area proportion of the Beijing Future Science City of China.

Land Use Type	Area (km^2^)	Percentage (%)
Land for public facilities	2.78	27.12
Multifunctional land use	0.45	4.44
Green land	3.45	33.71
Land for habitation	0.55	5.41
Road square land	1.89	18.48
Municipal public facilities	0.18	1.75
Water	0.93	9.08

**Table 3 ijerph-20-04640-t003:** Generalized statistics of the rainwater drainage system model of the Beijing Future Science City of China.

Division	Southern Area				Subtotal	Northern Area			Subtotal	Total
System (unit)	A	B	C	D	4	E	F	G	3	7
Number of nodes (per node)	18	58	11	67	154	38	6	6	50	204
Number of pipe segments (units)	17	57	10	66	150	37	5	5	47	197
Pipe length (m)	6410	12,110	2500	12,220	33,240	10,200	1870	990	13,060	46,300
Number of subcatchment areas	14	14	6	25	59	25	4	3	32	91
Control and discharge area (km^2^)	94.26	157.90	43.64	144.12	439.93	192.27	16.11	6.22	214.60	654.52

**Table 4 ijerph-20-04640-t004:** Urban comprehensive runoff coefficient values.

Area Type	Runoff Coefficient
Built-up dense area (impervious area: >70%)	0.60–0.80
Densely built area (impervious area: 50% to 70%)	0.50–0.70
Less built-up area (impervious area: 30% to 50%)	0.40–0.60
Very sparse building area (impervious area: <30%)	0.3–0.50

**Table 5 ijerph-20-04640-t005:** Hydraulic factors of circular pipes with different filling degrees.

Filling Degrees	*A*	*R*	Filling Degrees	*A*	*R*
0.05	0.0147 d^2^	0.0326 d	0.55	0.4422 d^2^	0.2649 d
0.1	0.04 d^2^	0.0635 d	0.6	0.492 d^2^	0.2776 d
0.15	0.0739 d^2^	0.0929 d	0.65	0.5404 d^2^	0.2881 d
0.2	0.1118 d^2^	0.1206 d	0.7	0.5872 d^2^	0.2962 d
0.25	0.1535 d^2^	0.1466 d	0.75	0.6319 d^2^	0.3017 d
0.3	0.1982 d^2^	0.1709 d	0.8	0.6736 d^2^	0.3042 d
0.35	0.245 d^2^	0.1935 d	0.85	0.7115 d^2^	0.3033 d
0.4	0.2934 d^2^	0.2142 d	0.9	0.7445 d^2^	0.298 d
0.45	0.3428 d^2^	0.2331 d	0.95	0.7707 d^2^	0.2865 d
0.5	0.3927 d^2^	0.25 d	1	0.7854 d^2^	0.25 d

Note: In fact, d has different values. For the convenience of expression, d is used in the table.

**Table 6 ijerph-20-04640-t006:** Designed rainfall in different rainfall return periods.

Return Period (Year)	10 Years	20 Years	50 Years	100 Years
Designed rainfall value (mm)	431.00	641.47	1147.78	1822.44
Accumulated surface water in the northern area (m^3^)	370,679.373	552,561.757	989,391.665	15,566,302.990
Accumulated surface water in the southern area (m^3^)	852,020.200	1,268,676.930	2,270,997.400	3,606,592.520

**Table 7 ijerph-20-04640-t007:** Waterlogging risk of different pipe filling degrees.

Pipe Filling Degrees *a*	Tube Flow State	Waterlogging Risk
*a* < 1	Open channel flow	None
1 < *a* < 5	Flow under Pressure	With risk but not high
5 < *a*	Flow under Pressure	High
10 < *a*	Flow under Pressure	Higher

**Table 8 ijerph-20-04640-t008:** Statistics of the comprehensive runoff coefficient of typical rainfall.

	Southern Area	Northern Area	Comprehensive Runoff Coefficient
Study Area (ha)	439.93	214.60	----
Impervious Area Ratio (%)	50%	45%	----
“21 July 2012” Heavy Rain	0.449	0.404	0.434

**Table 9 ijerph-20-04640-t009:** Adjustment results of the maximum flow of the pipe section connected to the drain port.

Outlet	A7–A8	B13–B14	C5–C6	F5–F6	G3–G4
Maximum filling degree of pipe section *a*	0.307	0.806	0.803	0.692	0.505
Pipe diameter *d* (m)	4 × 3	3.4 × 2.3	2.0	1.2	1.1
Cross-sectional area *A* (m^2^)	3.6840	6.3029	2.7035	0.8348	0.4812
Hydraulic radius *R* (m)	0.6306	0.8868	0.6083	0.3539	0.2766
Bottom slope (%)	0.23	0.06	0.06	0.08	0.06
Calculated value *Q_max_* (m^3^/s)	9.9941	10.9619	3.6571	0.9087	0.3849
Simulated value *Q_max_* (m^3^/s)	11.226	12.664	4.118	0.9890	0.4810
Relative error	12.33%	15.53%	12.60%	8.84%	24.96%

**Table 10 ijerph-20-04640-t010:** Calibrated and verified parameters of model.

	Parameter	Description	Value Range	Calibration Result
Total control parameter	Initial loss of rainfall (mm)	Initial water content of catchment	0.5–1.5	0.6
	Hydrologic attenuation coefficient	Impermeable ratio	0.6–0.9	0.9
Confluence control parameters	Average surface runoff velocity *v* (m/s)	Time required to travel from the farthest end of the basin to the outlet of the basin	0.25–0.30	0.3
	Manning coefficient of pipeline (*M* = 1/*n*)	Roughness of pipe	*M* = 5 − 75 (m^1/3^/*S*) or *n* = 0.009 − 0.017	75 or 0.013

**Table 11 ijerph-20-04640-t011:** Statistics of the simulation results of the influence of the designed rainfall process on the pipeline filling degree under different design return periods.

Return Period	Pipeline Filling Degree *a*	Northern Region		Southern Region	
		Number of Pipe Segments	Rainwater Well (Number of Associated Nodes)	Number of Pipe Segments	Rainwater Well (Number of Associated Nodes)
10 years	*a* < 1	3	4	20	20
	1 < *a* <5	45	45	130	114
	5 < *a*	1	1	13	10
	10 < *a*	0	0	1	1
20 years	*a* < 1	3	4	18	18
	1 < *a* < 5	43	41	125	111
	5 < *a*	6	5	14	14
	10 < *a*	0	0	2	2
50 years	*a* < 1	4	6	17	16
	1 < *a* < 5	18	15	131	109
	5 < *a*	26	23	17	19
	10 < *a*	6	6	1	1
100 years	*a* < 1	3	5	16	18
	1 < *a*< 5	16	9	138	105
	5 < *a*	37	28	18	20
	10 < *a*	6	8	2	2

## Data Availability

Not applicable.

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
