# Peer review of "Urban Flood Modeling and Risk Assessment with Limited Observation Data: The Beijing Future Science City of China"

_ijerph, 2023, doi:10.3390/ijerph20054640_

Round 1

Reviewer 1 Report

In this research, a new method of model calibration and verification is proposed, combining the three methods of empirical calibration, formula calibration, and survey calibration based on field investigation to build the drainage system model in the Beijing Future Science City of China. The results of the research are useful, but they are more suitable for the studied area, and it is not clear that they are applicable to other parts of the world. There is also a need to provide solutions for places that are flooded. Below are some comments to correct the manuscript:

1- Table 2 is out of order, and it is better to set it in a different form so that it can be analyzed better.

2- In Table 3, what is the basic range of parameters selected? Most of the parameters are in the maximum range and should be re-ranged.

3- The figures of research are small and unclear and need to be corrected.

4- In Eq 4,5: formulas of scope of applications must be checked. Also, formula parameters were not explained. 

5- Line 262: Thiess == Thiessen

Author Response

Dear Editor/Reviewers:

We appreciate you and the reviewers for your precious time in reviewing our paper "Urban flood modeling and risk assessment with limited observation data: The Beijing Future Science City of China" (ijerph-2227658) and providing valuable comments. It was your valuable and insightful comments that led to possible improvements in the current version. The authors have carefully considered the comments and tried our best to address every one of them. We hope the manuscript after careful revisions meet your high standards. The authors welcome further constructive comments if any. In the response letter, all the comments/suggestions are shown in black, all the responses are shown in red, and the modified results are highlighted in blue. In the manuscript, the modified contents are highlighted in yellow.

Best wishes,

All authors.

Reviewer 2 Report

For exploring the flooding risk of the Future Science and Technology Park in Beijing, China, the MIKE URBAN was selected to establish the urban waterlogging model in this study. Due to lack of the pipeline-flow observational data, the parameters of the model were calibrated using the empirical calibration method, and the reliability of the calibrated model was verified by the methods of formula verification and field survey verification. Finally, based on the established model, the regional waterlogging situations of design rainstorms in different return periods were analyzed and evaluated. This manuscript is very helpful for the establishment of the urban waterlogging model in areas without pipeline-flow observational data. The findings of this manuscript can provide guidance for regional sustainable development. However, some contents still need to be improved or revised before publication in this journal. The details are listed as follows:

1. Abstract: Since the emphasis of this study is on the calibration and verification of urban waterlogging model in areas without pipeline-flow observations combining the three methods, the quantitative results of calibration and verification need to be supplemented in the abstract.

2. Introduction: Please summarize the gaps of previous research and illustrate the innovation of this study.

3. Materials and methods: The logical structure of the "Materials and methods" section is confused. The focus of this section should be on the introduction of the study area, data sources, principles of the methods, and precipitation scenarios. The results of calibration and verification should be presented in the "Results" section.

4. All regions: Since the northern and southern regions are independent from each other, there is no water exchange and interaction. Is it meaningful to compare the differences between the northern and southern regions when analyzing the waterlogging risk across the region?

5. Discussion and conclusion: It is recommended that the main results and findings of this study be expressed quantitatively in the conclusion section.

6. The author needs to revise and standardize all the Figures in the manuscript. For example, Figure 1 is short of necessary elements. Figure 4 and Figure 5 style is not uniform. The design rainstorms processes are depicted in Figure 5, however specific dates exist.

7. There are many minor issues in the manuscript that need to be revised. For example, the references citation in the manuscript needs to be standardized. English language expression needs to be standardized. Formulas need to be written in a standardized way. The author needs to carefully proofread the full manuscript in the next round.

Author Response

(The authors gave the same response as above.)

Reviewer 3 Report

This manuscript (ijerph-2227658) aims to apply the commonly-used MIKE URBAN method to build the drainage system model in the Beijing Future Science City of China. Although it is an easy-to-follow manuscript, it is not entirely new to use these normal methods in urban flood risk assessment. Another very serious concern is that some related studies have been neglected. A further and detailed literature review must be conducted. Also, the current results of this study can hardly be reviewed because of those problems about data and methodology. Therefore, a “Major Revision” is required. My suggestions and comments are presented as follows:

- 1. First of all, the overall structure of the paper needs to be deepened and the significance and innovation of the research needs to be further demonstrated.

- 2. Both the Abstract and the Introduction Section are a bit weak because the authors did not clearly raise an important scientific question or gap related to urban flood risk assessment. Therefore, potential readers can hardly identify the need that the authors should have to provide a new solution from an international perspective. What I have learned from the introduction is that the authors applied some previous established models (e.g., MIKE URBAN) to some specific study areas (Beijing Future Science City of China). Note that those methods are not new methods or concepts in urban flood risk assessment.

- 3. In the Introduction, the authors have devoted too much space on the description and review of the SWMM model, which should be shorten and condensed because this study did not really select or use the SWMM model.

- 4. In the middle of the Introduction, the authors have mentioned a number of studies related to urban flood risk assessment, but without mentioning the disadvantages of these studies.

- 5. Actually, in Line 50 ~ 52: the authors have mentioned that: "Knowing how to conduct model simulations in urban areas that lack monitoring data has become a difficult problem in storm flood risk management". However, some other methods have also been developed to deal with these kinds of problems, which has been neglected in the current manuscript.

- 6. Another serious concern is that the authors must look further into the latest research in this field. In fact, the literature review is far from enough. In particular, the maximum entropy algorithm has been successfully used in urban flood (waterlogging) risk assessment. However, this well-accepted technique is ignored in the manuscript, and the following articles should be mentioned. The Introduction section is meant to set the context for your research work and highlight how it contributes to the knowledge in this field and builds on previous similar studies.

Predicting future urban waterlogging-prone areas by coupling the maximum entropy and FLUS model. Sustainable Cities and Society, 2022, 80: 103812

Utilizing User-Generated Content and GIS for Flood Susceptibility Modeling in Mountainous Areas: A Case Study of Jian City in China. Sustainability, 2021.

Hydrometeorological disaster risk assessment in upper Gori–Ramganga catchment, Uttarakhand, India. Geocarto International, 2022.

- 7. In Section 2, please provide a new figure showing the overall roadmap of the entire research.

- 8. In Section 2.1. Study Area: the authors need to explain why they selected such a small study area (with an area of just 10 km2) for model examination.

- 9. The authors failed to provide many specific details of the input data, such as the pre-processing processes, date, resolution, and accuracies. I suggest the authors to provide this information in a new table.

- 10. Table 4. Urban comprehensive runoff coefficient values: the authors need to explain more clearly how to calculate these runoff coefficient values.

- 11. The Discussion and Conclusion Section should be separated into two different sections.

- 12. Discussions: Basically, there is no discussion. The Discussion and Conclusion Sections failed to engage with the wider readership of this international journal. For example, most contents in the abstract are only related to the specific study area. The novelty and originality should be clearly justified that the manuscript contains sufficient contributions to the new body of knowledge from the international perspective.

- 13. The authors also need to improve the Conclusion Section by mentioning the main shortages of your work.

- 14. There are too many figures and tables in the current manuscript. Actually, some figures and tables are unnecessary because the information is well known to the potential readers and researchers in this journal.

Author Response

(The authors gave the same response as above.)

Round 2

Reviewer 1 Report

Thanks to the authors.

Reviewer 3 Report

Thank you for incorporating my comments and suggestions.